# Precision-edited histone tails disrupt polycistronic gene expression controls in trypanosomes

Markéta Novotná ®[1], Michele Tinti ®[1], Joana R. C. Faria ®[2] & David Horn ®[1]✉

Transcription of protein coding genes in trypanosomatids is atypical and almost exclusively polycistronic. In *Trypanosoma brucei*, for example, approximately 150 polycistrons, and 8000 genes, are constitutively transcribed by RNA polymerase II. The RNA pol-II promoters are also unconventional and characterised by regions of chromatin enriched for histones with specific patterns of post-translational modification on their divergent N-terminal tails. To investigate the roles of histone tail-residues in gene expression control in *T. brucei*, we engineered strains exclusively expressing mutant histones. We used an inducible CRISPR-Cas9 system to delete >40 histone *H4* genes, complementing the defect with a single ectopic *H4* gene. The resulting "hist^one^H4" strains were validated using whole-genome sequencing and transcriptome analysis. We then performed saturation mutagenesis of six histone H4 N-terminal tail lysine residues, that are either acetylated or methylated, and profiled relative fitness of 384 distinct precision-edited mutants. H4$^{lys10}$ mutations were not tolerated, but we derived nineteen strains exclusively expressing distinct H4$^{lys4}$ or H4$^{lys14}$ mutants. Proteomic and transcriptomic analysis of H4$^{lys4}$ glutamine mutants revealed significantly reduced expression of genes adjacent to RNA pol-II promoters, where glutamine mimics abnormally elevated acetylation. Thus, we present direct evidence for polycistronic expression control by modified histone H4 N-terminal tail residues in trypanosomes.

Trypanosomatids are excavate protozoa that include several flagellated and vector-transmitted parasites[1]. The African trypanosome, *Trypanosoma brucei*, for example, is transmitted by tsetse flies and causes lethal human and animal diseases. Trypanosomatids display several unusual features for eukaryotes, one of which is a genome organised into very long, constitutively active, polycistronic, RNA polymerase II transcription units, comprising many genes with unrelated functions[2]. *T. brucei* also employs RNA polymerase I to transcribe both *rRNA* and some protein-coding genes, including sub-telomeric variant surface glycoprotein (VSG) genes, which are expressed in a monoallelic fashion in bloodstream-form cells[3,4]. Trypanosomatid histones display substantial differences relative to other model eukaryotes, and impacts on the *T. brucei* octameric nucleosome core particle, comprising approximately 146 bp of DNA wrapped around two molecules each of histones H2A, H2B, H3, and H4, have recently been elucidated using cryo-EM for structural analysis[5].

Histones, histone tails that extend beyond the core nucleosome, histone post-translational modifications, and histone variants can impact gene expression. Histone N-terminal tails in particular display disordered and exposed surfaces that may recruit regulatory factors. Indeed, modifications, added by enzymes called writers, removed by erasers, and interpreted by proteins with reader domains, have been

[1]School of Life Sciences, University of Dundee, Dundee, UK. [2]Biology Department and York Biomedical Research Institute, University of York, York, UK. ✉e-mail: d.horn@dundee.ac.uk

investigated in *T. brucei*. For example, histone modifications have been mapped using mass spectrometry[6–8], while the roles of histone writers, erasers and readers have been investigated following knockdown[6], by chromatin immunoprecipitation, and by subcellular localisation[9]. In a few cases, *T. brucei* writers have been linked to particular histone substrates and post-translational modifications. The histone acetyltransferase, HAT2 acetylates histone H4 at the lysine 10 position, for example[6,10], while HAT3 acetylates histone H4 at the lysine 4 position[11].

Although the long polycistrons in trypanosomatids are typically constitutively transcribed by RNA polymerase II, locus-specific regulation has been reported in both *Leishmania*[12] and *T. brucei*[13]. In *T. brucei*, the regions that recruit RNA polymerase II to initiate transcription contain GT-rich tracts and display characteristics of highly dispersed or distributed promoters that otherwise lack conventional promoter-associated sequences[14,15]. These unconventional promoters are characterised by regions of 5–10 kbp that are enriched for a large number of putative chromatin regulatory factors[9], variant histones H2A.Z and H2B.V, and more than fifty histone modifications; histone H4 acetylated at the lysine 2 and 10 positions, for example[6,16]. Transcription termination sites (TTS), on the other hand, are enriched for histone variants H3.V and H4.V, and for a DNA base J modification[13,16]. In terms of the three-dimensional organisation of the nucleus, RNA polymerase II transcription initiation sites cluster in inter-chromosomal transcription hubs[17], while a single active *VSG* is associated with an inter-chromosomal RNA polymerase I transcription and *trans*-splicing compartment[18].

It can be challenging to establish the precise role of a particular histone post-translational modification. In the case of transcription, for example, accumulation of histone modifications and chromatin-associated factors may cause transcriptional changes or be a consequence of those changes. Further complicating interpretations, histone writers, readers, and erasers often operate as components of multi-subunit complexes, and often have multiple, and combinatorial histone and non-histone substrates. Homogeneous mutant genotypes can be generated to either prevent or mimic specific histone post-translational modifications, and this was first achieved in *Saccharomyces cerevisiae*, which has only two copies of each core histone gene; early studies in this yeast revealed surprising and substantial redundancy among histone N-terminal tails[19,20]. The generation of homogeneous mutant histone genotypes is typically challenging in other species, however. This required the expression of twelve copies of mutant histone *H4* in *Drosophila*, for example[21]. Analysis of these strains illustrated the benefits of assessing mutant histones directly, revealing that a histone H4 lysine 20 methylation defect was not associated with the range of expected gene expression and developmental phenotypes, counter to prior interpretations based on studies in both humans and flies[22]. Similarly, analysis of base-edited mouse cells, counter to prior suggestions, revealed that H3 lysine 27 acetylation was not required for gene derepression; the adenine base editor used in this case also edited some adjacent serine 28 residues[23].

Interpretations from studies involving writer, reader or eraser perturbation are similarly challenging in trypanosomatids, which have more than 150 distinct histone modifications[6]. Transcription was perturbed following depletion of *T. brucei* histone acetyltransferase 2 (HAT2), for example, but acetylation was significantly reduced at histone H4 lysines 2, 5 and 10, as well as acetylation at multiple sites on the histone variants, H2A.Z and H2B.V[6]. Similarly, histone H4 lysine 4 acetylation is reduced but is not eliminated in *hat3* null cells, indicating acetylation by more than one acetyltransferase[24]. In *T. brucei*, all four core histone genes are present in tandem arrays, with approximately 43 copies of the histone *H4* gene, for example. It was possible, using a CRISPR-Cas9 system over five months to edit many of these *H4* genes, but 10% remained unedited and 40% were deleted[25]. Using an alternative approach, myc-tagged and mutant histone *H4* genes were inducibly expressed in *Trypanosoma cruzi*, but these histones were

estimated to contribute only 0.2% or 1.4% of the total histone H4 pool in these cells[26].

We have engineered *T. brucei* strains that express a single ectopic *H4* gene and that lack the histone *H4* tandem arrays. These hist^one^H4 strains provide a platform for generating and analysing homogeneous mutant histone *H4* genotypes. Accordingly, precision site-saturation mutagenesis was used to generate and assess hundreds of histone H4 N-terminal tail lysine mutants. Analysis of strains exclusively expressing mutant histones provided direct evidence for polycistronic gene expression control by H4 N-terminal tails.

## Results

### Replacement of >40 native *T. brucei* histone *H4* genes

Replication-dependent core histones are encoded by tandem gene arrays in *T. brucei*. Specifically, in the case of the histone *H4* genes, there are estimated to be 43 copies in the diploid genome in two arrays, each of approximately 15 kbp in length[25]. To facilitate *H4* gene editing, we generated *T. brucei* strains with a single copy of the histone *H4* gene. We first integrated a cassette that incorporated an ectopic $H4^{ECT}$ gene and that encoded alternative $H4^{NAT}A$ or $H4^{NAT}B$ single guide RNAs (sgRNAs) in the $2T1^{T7-Cas9}$ strain[27], which inducibly expresses Cas9 (Fig. 1a). Both sgRNAs targeted the native *H4* arrays, but not the $H4^{ECT}$ gene, which encoded the same 100 amino acid protein but was recoded with 53 synonymous changes (Fig. 1a) such that every third codon position is a G or a C, which favours increased expression[28,29]. The synonymous changes additionally eliminated protospacer adjacent motifs associated with the sgRNAs, and altered 2–4 bases within each sgRNA target sequence, protecting the $H4^{ECT}$ gene from Cas9-targeting. The $H4^{ECT}$ gene was flanked by native *H4* untranslated and *cis*-regulatory regions thought to support S phase specific expression[30], and each expression cassette also incorporated a T7 phage promoter to drive transcription by the T7 polymerase, also expressed in the $2T1^{T7-Cas9}$ strain[27]; T7 polymerase achieves approximately 40-fold higher expression relative to RNA polymerase II in *T. brucei*[31], and can drive expression of protein-coding genes in trypanosomatids because mRNA capping is achieved via *trans*-splicing. The system was, therefore, designed to replicate native *H4* expression, and to complement for the loss of >40 native *H4* genes with a single recoded $H4^{ECT}$ gene.

Taking strains expressing $H4^{ECT}$ and either the $H4^{NAT}A$ or $H4^{NAT}B$ sgRNA, we induced Cas9 expression, and delivered a *Neomycin PhosphoTransferase* (*NPT*) cassette, designed to replace the native *H4* arrays, 24 h later (Fig. 1a). Screening of transformed clones using a PCR-assay revealed correct integration of the *NPT* cassette, driven by the $H4^{NAT}B$ sgRNA, but not by the $H4^{NAT}A$ sgRNA (Supplementary Fig. 1a). A second PCR assay suggested that both histone *H4* arrays had been replaced in one of three "H4^NATB" clones (Supplementary Fig. 1b). We repeated the native *H4* deletion process using the $H4^{NAT}B$ strain, screened a second panel of clones, and isolated a second independent hist^one^H4 clone. Both hist^one^H4 strains were further validated using Southern blotting (Fig. 1b) and whole genome sequencing (Fig. 1c). Both approaches confirmed deletion of all native *H4* genes in the hist^one^H4 strains. Genome sequencing also revealed that removal of the native *H4* gene arrays on chr. 5 in the hist^one^H4 strains was both precise and specific (Fig. 1c).

### A recoded *H4* gene complements for the loss of native *H4* genes

We next characterised the hist^one^H4 strains to assess complementation for the loss of >40 native *H4* genes by the single ectopic $H4^{ECT}$ gene. We first demonstrated similar growth rates for the $2T1^{T7-Cas9}$ and hist^one^H4 strains (Supplementary Fig. 1c). A Reverse-Transcription PCR assay was then used to detect transcripts from both $H4^{ECT}$ and native *H4* genes, suggesting exclusive expression of the ectopic $H4^{ECT}$ gene in both hist^one^H4 strains, as expected (Supplementary Fig. 1d).

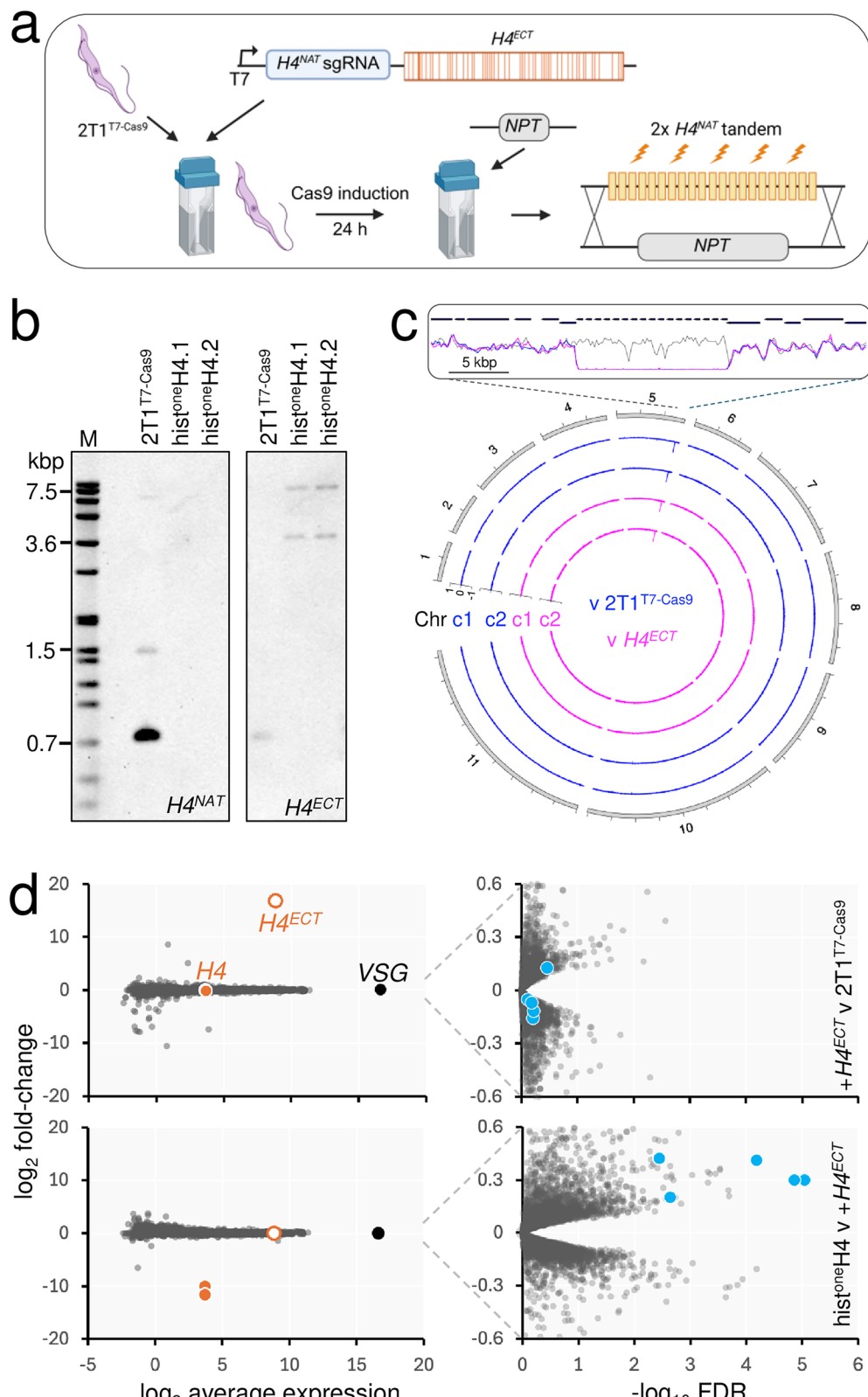

We next progressed to RNA-seq analysis of parental 2T1$^{T7\text{-}Cas9}$ cells, cells additionally expressing the $H4^{ECT}$ gene, and hist$^{one}$H4 cells lacking the native $H4$ genes. $H4^{ECT}$ gene recoding meant that we could readily distinguish between native $H4$ and $H4^{ECT}$ expression and indeed, the transcriptomes revealed highly significant changes in the expression of these two genes. Specifically, the $H4^{ECT}$ gene registered robust expression in the presence or absence of the native $H4$ genes (Fig. 1d, left-hand panels; log$_2$CPM = 8.9, 80+/−1% relative to total native $H4$ transcripts). Additionally, the native $H4$ signal was depleted >1000-fold following $H4$ array deletion, as expected (Fig. 1d, left-hand panels). A closer inspection of genes that registered significantly different expression highlighted 20−42% increased expression of five adjacent genes immediately downstream of the $H4$ arrays, specifically following deletion of the $H4$ arrays (Fig. 1d, right-hand panels). We speculate that

**Fig. 1 | Complementation of >40 *T. brucei* histone *H4* genes with a recoded *H4* gene. a** The schematic shows hist[one]H4 strain construction. The 2T1[T7-Cas9] strain was transfected with a cassette containing a recoded, and ectopic histone *H4* gene (*H4[ECT]*, synonymous changes highlighted) and a single guide RNA (sgRNA) targeting *H4[NAT]* genes, both under the control of a T7 RNA polymerase promoter (T7). Cas9 was induced for 24 h in the resulting cell line, which was then transfected with an *NPT* (neomycin phosphotransferase) cassette to delete the native histone *H4* (*H4[NAT]*) tandem arrays. Created in BioRender. Novotna, M. (2025) https://BioRender.com/f0do2k7. **b** The Southern blots were probed for the 5′ end of the native *H4* genes (left), which detected the expected 736 bp band (approx. 43 copies), and 1505 bp band (1 copy from each of 2 alleles) only in the parental cell line. The blot was stripped and re-probed for the ectopic *H4* gene (right) revealing the expected 4664 bp and 13803 bp bands in both independent hist[one]H4 strains,

but not in the parent. Some residual signal, or cross-reactivity, can be seen from the 736 bp band; the ectopic *H4* probe shares 84% sequence identity with native *H4* genes. M, digoxigenin (DIG)-labelled DNA ladder. **c** The circular plot shows whole-genome sequencing data for the hist[one]H4 strains compared to the parental strain, and the parental strain plus the *H4[ECT]* gene. The zoom at top shows precise deletion of the native *H4* arrays; clone 1, blue; clone 2, magenta. **d** RNA-seq analysis. A strain expressing the *H4[ECT]* gene was compared to the parent strain in the upper panels. Both hist[one]H4 strains were compared to the strain expressing the *H4[ECT]* gene in the lower panels. The *H4[NAT]* and *H4[ECT]* transcripts (orange) and the super-abundant *VSG* transcript are highlighted in the left-hand panels, while five genes immediately downstream of the *H4* array are highlighted (blue) in the right-hand panels. *n* = 8934.

this may be due to a post-transcriptional increase in mRNA maturation, since histone gene arrays have been shown to associate with *trans*-splicing loci in chromosome conformation capture assays[18]. We concluded that the single ectopic *H4[ECT]* gene complemented for the precise deletion of >40 native *H4* genes in two independent hist[one]H4 strains.

## H4 tail lysine saturation mutagenesis and multiplex fitness-profiling

While human and *S. cerevisiae* histone H4 N-terminal tail sequences differ at only V[21]I, there are substantial differences in *T. brucei* (Fig. 2a). The *T. brucei* sequence, like the sequences in human and *S. cerevisiae*, is lysine (K)-rich, however; seven of the first eighteen residues are K in *T. brucei*. To enable *H4[ECT]* gene editing, we replaced the H4[NAT]B sgRNA in both hist[one]H4 strains with an sgRNA that reprogramed Cas9 to target the *H4[ECT]* gene (Fig. 2b). To facilitate editing of K residues in the H4[ECT] N-terminal tail, we used an sgRNA that guided the introduction of a break 41 bp from the ATG start-codon. Hist[one]H4 strains, now also incorporating the *H4[ECT]* sgRNA, were validated by PCR and Sanger sequencing and two independent clones were selected for editing.

We designed editing templates for marker-free site saturation mutagenesis at H4[K4], H4[K10] and H4[K14]; lysine residues that are known to be acetylated and/or methylated[6,7], that have been characterised in trypanosomes[6,10,11,26], and that are also conserved in *Leishmania* histone H4. Each single-stranded DNA template incorporated a randomised codon and several synonymous bases, flanked by 25 b homology arms (Supplementary Fig. 2a and Supplementary Data 1). To protect edited *H4[ECT]* genes from further Cas9-targeting, the synonymous bases both eliminated the targeted protospacer adjacent motif and altered bases within the sgRNA sequence. The introduction of synonymous bases also served a second purpose here, by generating novel PCR primer binding sites to facilitate the amplification of edited sequences. Taking both hist[one]H4 strains expressing the *H4[ECT]* sgRNA, we induced Cas9 expression and individually delivered the three editing templates, each with a distinct randomised codon, 24 h later; thereby yielding two independent *T. brucei* libraries for each edited lysine residue. Samples were collected for DNA extraction before template delivery, and from all six libraries 0.5, 2, 4 and 6 days after template delivery (Fig. 2b), with Cas9 induction maintained throughout.

To profile the edits, we selectively amplified edited sequences and also amplified unedited sequences in parallel (Supplementary Fig. 2b). PCR products for unedited sequences were obtained from all thirty samples. In contrast, and as expected, edited products were only detected in those twenty-four samples taken after delivery of an editing template, both confirming editing and demonstrating specific amplification of edited *H4[ECT]* sequences. We deep-sequenced the multiplexed and edited amplicons to profile the relative fitness of cells harbouring each of the possible edits; 64 codons at either H4[K4], H4[K10] or H4[K14]. We then scored sequence-reads for all possible codons at each position and at each time-point and visualised the outputs on

radial plots (Fig. 2c, upper panels); data from the duplicate libraries were pooled to generate the radial plots since these replica experiments yielded highly consistent results based on principal component analysis (Supplementary Fig. 3a). By comparing the 2-day samples to the 0.5-day baseline, we were able to confirm the introduction of all 192 possible edits (Fig. 2c, upper panels, blue inner datasets). Notably, the three stop codons were the most rapidly depleted edits, regardless of the K-residue edited, consistent with the expected loss-of-function phenotypes in strains expressing a single *H4* gene with a nonsense mutation.

We next turned our attention to assessing the relative fitness of cells with mis-sense mutations. Samples across the 6-day time-course revealed broadly consistent trends for synonymous edits that encode common amino acids (Fig. 2c, upper panels). The analysis revealed that H4[K10] mutations were not tolerated; although all mis-sense edits at the K[10] position were similarly represented in the day-2 samples, all but the pair of lysine codons were severely diminished in the day-4 and day-6 samples (Fig. 2c, upper middle radial plot). These results suggested a severe growth defect regardless of the alternative amino acid at this position, and an important role for H4[K10]. H4[K4], in contrast, can be replaced by positively charged (histidine, H; or arginine, R) or aromatic (phenylalanine, F; or tyrosine, Y) amino acids, while codons representing negatively charged amino acids (aspartic acid, D; or glutamic acid, E) were rapidly depleted (Fig. 2c, upper left-hand radial plot). K[14] can also be replaced, but primarily with non-polar residues in this case (isoleucine, I; methionine, M; valine, V; alanine, A; or glycine, G), while codons representing the aromatic amino acids (phenylalanine, F; tryptophan, W; or tyrosine, Y) were rapidly depleted (Fig. 2c, upper right-hand radial plot).

To extend and further test the utility of the hist[one]H4 strains and the platform we developed for assessing fitness associated with homogeneous histone H4 mutant genotypes (Fig. 2b), we designed three further editing templates and implemented site saturation mutagenesis at H4[K2], H4[K17] and H4[K18]. These lysine residues are also known to be acetylated and/or methylated[6,7], and are conserved in *Leishmania*; H4[K5] is not conserved in *Leishmania*. We scored sequence-reads for all possible codons at each position and at each time-point and once again observed highly consistent results from the replica experiments (Supplementary Fig. 3b). Pooled datasets were visualised on radial plots (Fig. 2c, lower panels) and once again confirmed the introduction of all possible edits and rapid depletion of all three nonsense mutations, regardless of the K-residue edited (Fig. 2c, lower panels, blue inner datasets). Samples across the 6-day time-course, once again revealed broadly consistent trends for individual amino acids (Fig. 2c, lower panels). Nonpolar residues were tolerated at the K[2] position, as also seen for K[14] above. The K[17] position appeared to show some tolerance for polar residues, while mis-sense edits were typically not well-tolerated at the K[18] position. The results for each of the six edited histone H4 tail lysines are summarised as sequence logos, ranking those edits that were tolerated and those that were not tolerated (Fig. 2d). Only lysine registered as tolerated at all six sites, while

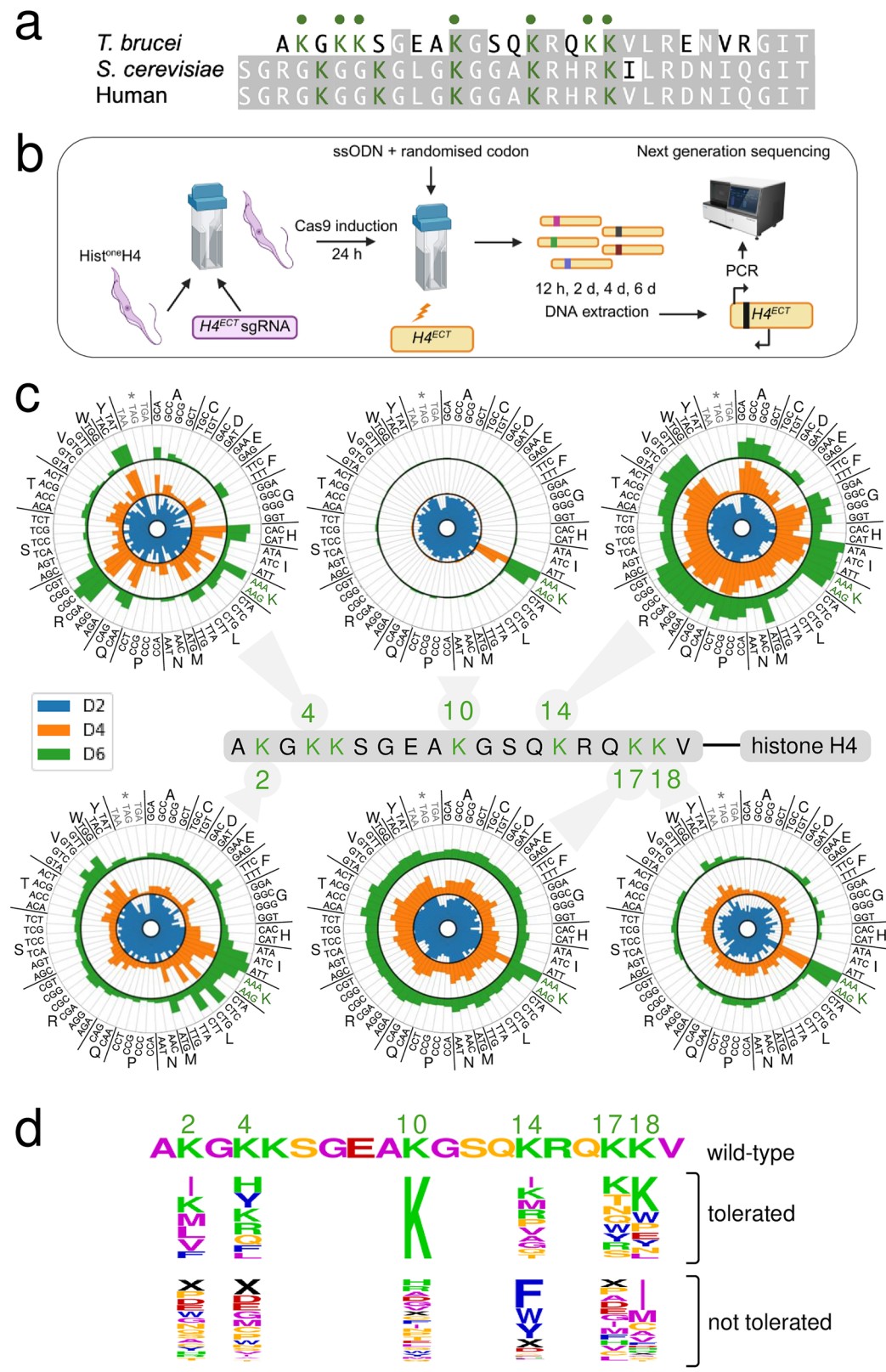

only nonsense codons (X), cysteine (C) and aspartic acid (D) codons registered as not tolerated at any of the six sites.

**A panel of strains exclusively expressing histone H4 mutants**

The analyses above revealed those histone N-terminal tail lysine edits that were tolerated at the K⁴ or K¹⁴ positions, but did not directly demonstrate the viability of strains homogeneously expressing these edited histones. To isolate individual histone H4 mutants, saturation mutated H4^K4, H4^K10 or H4^K14 cultures were subcloned two to ten days after mutagenesis. Clones were assessed using the PCR-assays described above to amplify unedited and edited sequences, and the latter amplicons from those clones that registered as negative and positive, respectively, were sequenced. Two hundred and twenty-six clones were assessed in total, 113 following K4 editing (14% edited), 62

**Fig. 2 | H4 tail lysine saturation mutagenesis and multiplex fitness-profiling. a** The alignment shows the N-terminal tail sequence of *T. brucei* histone H4, compared to the equivalent sequences from yeast and human. Lysine residues are highlighted in the *T. brucei* sequence. **b** Schematic of the saturation mutagenesis and amplicon-sequencing strategy. An sgRNA cassette targeting the 5′ end of the *H4ECT* gene was introduced into an hist^one H4 strain. Cas9 was induced, and 24 h later the cells were transfected with single-stranded oligodeoxynucleotide (ssODN) editing templates. DNA was extracted at different timepoints, the edited region of the *H4ECT* gene was PCR-amplified, and the amplicons were deep-sequenced.

Created in BioRender. Novotna, M. (2025) https://BioRender.com/21f7d0g. **c** The radial plots show the abundance of sequence-reads representing each codon at each edited position during the time course. Data for six edited N-terminal tail lysine's are shown. Values represent the averages of duplicate samples and are relative to codon scores obtained at the 12 h timepoint. D2, day-2; D4, day-4; D6, day-6. *, stop codons. Source data are provided as a Source Data file. **d** The logo shows relative overrepresentation or underrepresentation at day-6 for those cognate amino acids that were either tolerated or not tolerated following editing at each of the six lysine positions targeted.

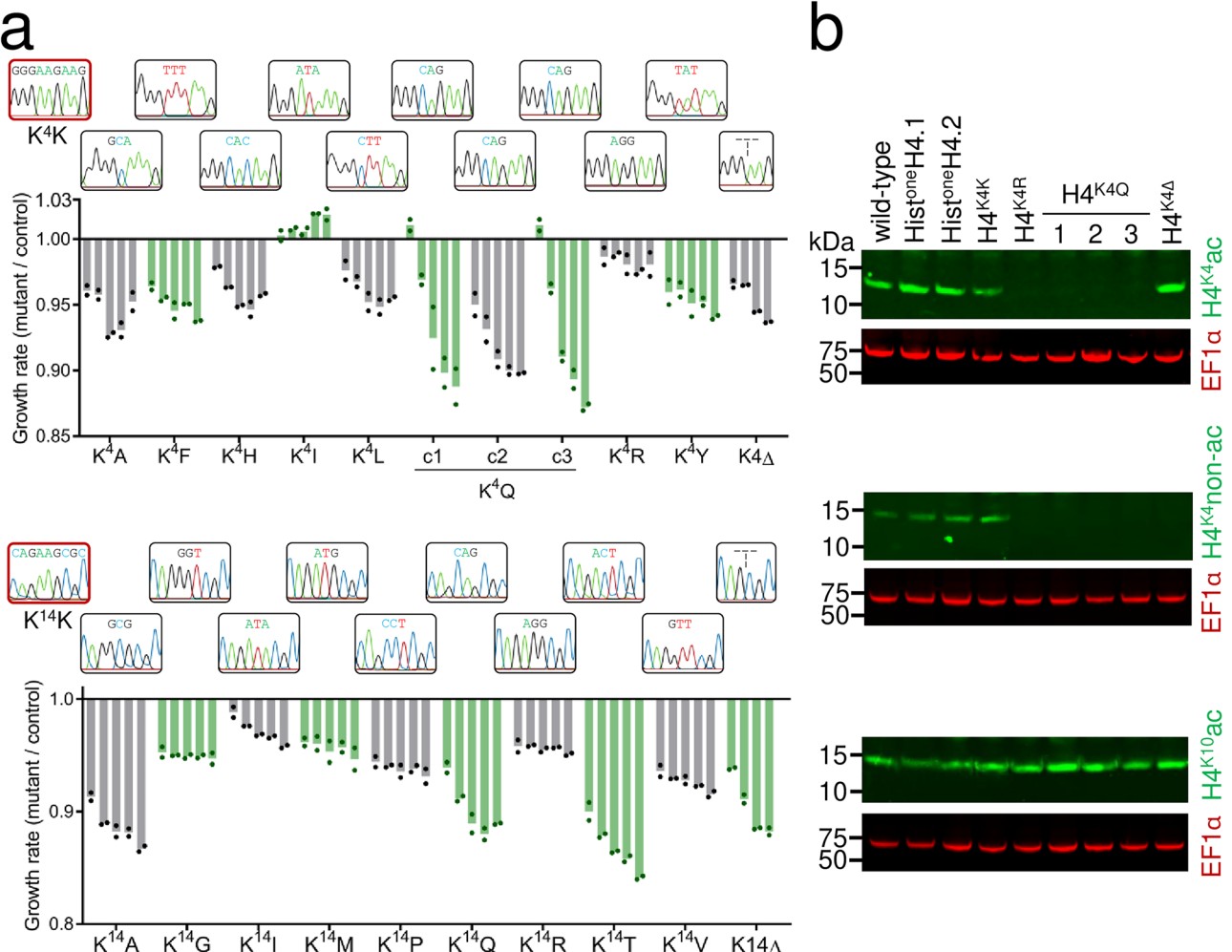

**Fig. 3 | A panel of strains exclusively expressing histone H4 mutants. a** The plots show the fitness of a panel of K4 and K14 mutants relative to H4^K4K or H4^K14K strains, respectively. Cell density was recorded every 24 h for 4–5 days (left to right) and for two technical replicates. Sanger sequencing traces show the edited codon for each mutant. **b** Protein blotting analysis of selected K4 mutants, showing H4^K4 acetylation, non-acetylated H4^K4, and H4^K10 acetylation. EF-1α (elongation factor 1 α; red) served as a loading control. Similar results were obtained from at least two independent experiments. Source data are provided as a Source Data file.

following K10 editing (8% edited), and 41 following K14 editing (95% edited), yielding strains exclusively expressing H4^K4K, H4^K10K or H4^K14K and a range of distinct H4^K4 or H4^K14 mutants (Fig. 3a). The panel of mutants we isolated was highly predictable, based on the fitness profiles described above (Fig. 2c, d); A, F, I, L, R or Y replaced K4, only K "replaced" K10, and A, G, I, M, P, R, T or V replaced K14.

Although K4Q (glutamine), K14Q, and K4H mutants were not obtained using the approach above, the fitness profiles (Fig. 2c, d) suggested that these mutants would be viable. We also wondered whether cells lacking the K4 or K14 residues would be viable. To obtain K4Q, K4Δ, K14Q, K14Δ and K4H mutants, we performed mutagenesis of

these residues as above (Fig. 2b), but in this case using templates containing specific codons or codon deletions rather than randomised codons. Editing, subcloning and assessment as above, yielded all five of these mutants; we screened twelve sub-clones in each case. The full panel of nineteen mutants, plus two additional and independent K4Q mutants, since we were particularly interested in this edit (see below), were assessed for relative fitness by directly measuring growth, and compared to H4^K4K or H4^K14K cells, as appropriate (Fig. 3a). Relative fitness of the K4 and K14 mutants were broadly predictable, again consistent with the fitness profiles described above (Fig. 2c, d). Most mutations were well-tolerated, resulting in <10% reduced growth over

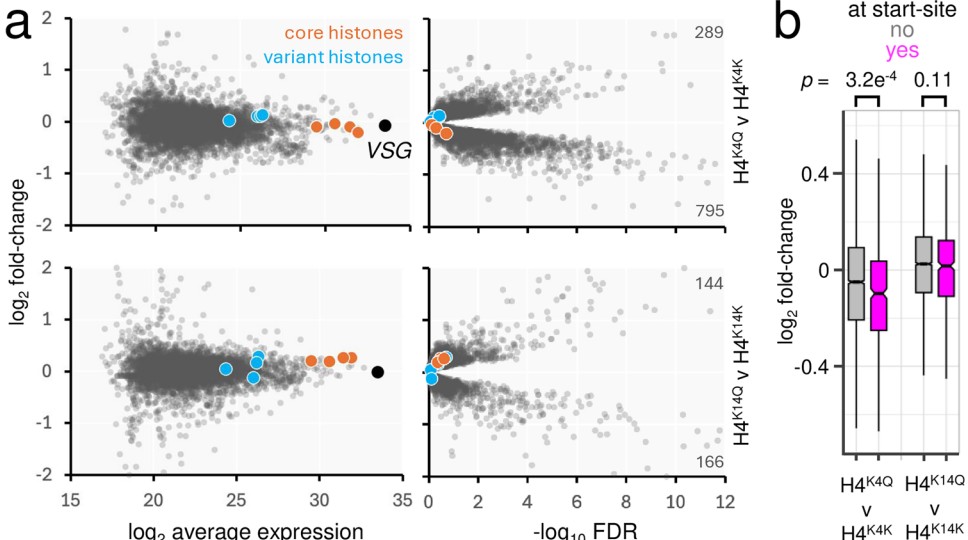

**Fig. 4 | Proteomic analysis of histone H4$^{K4Q}$ and H4$^{K14Q}$ mutants. a** Proteomics analysis. Three strains expressing the H4$^{K4Q}$ mutant were compared to an H4$^{K4K}$ control in the upper panels. A strain expressing the H4$^{K4IQ}$ mutant was compared to an H4$^{K14K}$ control in the lower panels. The core and variant histones are highlighted. $n = 5776$. **b** The boxplot shows log$_2$ fold-change for the two comparisons in (**a**), and

for either genes within 10 kbp of transcription start-sites ($n = 468$) or >10 kbp distal from those sites ($n = 5308$). Boxes indicate the interquartile range (IQR), the whiskers show the range of values within $1.5 \times$ IQR and a horizontal line indicates the median. The notches represent the 95% confidence interval for each median. $p$-values were calculated using two-sided $t$-tests.

4–5 days (Fig. 3a). The greatest loss of fitness among the K4 mutants was for the K$^4$Q mutants. K4 deletion was also tolerated, although this mutant is in fact equivalent to a K$^5\Delta$ mutant, in that it retains a K residue at the fourth position (see Fig. 2a).

Substitution of lysine with arginine (R) or glutamine (Q) serves to mimic non-acetylated or constitutively acetylated lysine, respectively. We observed a greater fitness cost associated with both the H4$^{K4Q}$ or H4$^{K14Q}$ mutants relative to the H4$^{K4R}$ or H4$^{K14R}$ mutants (Fig. 3a), suggesting that mimicking constitutive acetylation at these sites is more deleterious than blocking acetylation. To further assess and validate the H4$^{K4}$ mutant strains, we used protein blotting with a series of H4-tail modification-specific antibodies, that identify acetylated or non-acetylated H4$^{K4}$ [11] or acetylated H4$^{K10}$ [10]. Acetylated and non-acetylated H4$^{K4}$ were detected in wild-type cells, in hist$^{one}$H4 cells and in edited H4$^{K4K}$ cells, as expected (Fig. 3b; top and middle panels). In contrast, and also as expected, neither antibody recognised edited H4$^{K4R}$ or H4$^{K4Q}$ histones; although Q mimics acetylated lysine, this modification is not recognised by anti-H4$^{K4}$ac. Notably, anti-H4$^{K4}$ac did recognise edited K$^4\Delta$ histones, consistent with the view that this mutant does in fact retain a K residue at the fourth position. Finally, anti-H4$^{K10}$ac recognised H4 in all of these strains, indicating that these mutations do not substantially disrupt acetylation at this site, shown above to play an important role in terms of maintaining viability (Fig. 2c, d). These results confirm homogeneous expression of mutant histones in the H4$^{K4Q}$, H4$^{K4R}$ and H4$^{K4\Delta}$ strains.

### Proteomic analysis of histone H4$^{K4Q}$ and H4$^{K14Q}$ mutants

The finding that both H4$^{K4Q}$ and H4$^{K4R}$ mutants were viable suggested that dynamic H4$^{K4}$ acetylation is not required for viability and presented an opportunity to explore how histone H4 tail lysine residues impact gene expression. We were particularly interested in the H4$^{K4Q}$ mutants because H4$^{K4}$ acetylation is not enriched in regions where histone H4$^{K10}$ acetylation demarcates RNA polymerase II promoters; indeed this mark is diminished at these sites[6]. We initially used data-independent acquisition mass spectrometry to examine the proteomes of H4$^{K4K}$, H4$^{K4Q}$, H4$^{K14K}$ and H4$^{K14Q}$ mutants. We analysed three independent H4$^{K4Q}$ mutants, the H4$^{K4K}$ mutant served as an otherwise

isogenic control, and the H4$^{K14K}$ and H4$^{K14Q}$ mutants allowed us to assess the impact of a K-Q mutation at a different site. We first assessed the abundance of the core histones and the variant histones, H2A.Z, H2B.V, H3.V and H4.V, none of which were significantly different in abundance in the H4$^{K4Q}$ or H4$^{K14Q}$ mutants, relative to their H4$^{K4K}$ or H4$^{K14K}$ counterparts (Fig. 4a; all with -log$_{10}$ FDR < 1). This analysis also revealed that the variant histones were approximately 30-fold less abundant on average that the core histones. Further comparison of the proteomes of H4$^{K4K}$ and H4$^{K4Q}$ strains (Fig. 4a, top panels) and H4$^{K14K}$ and H4$^{K14Q}$ strains (Fig. 4a, bottom panels) revealed more substantial differences in the H4$^{K4Q}$ cells, relative to their H4$^{K4K}$ counterparts. Considering a -log$_{10}$ FDR threshold of >2, 289 and 795 proteins were significantly increased or decreased in abundance, respectively in these mutants, while 144 and 166 proteins were significantly increased or decreased in abundance in the H4$^{K14Q}$ mutants. Thus, the abundance of 3.5 times more proteins was significantly different in the H4$^{K4Q}$ mutants and 73% of these were significantly decreased in abundance.

To ask whether changes in expression were associated with gene location within polycistronic transcription units, we assessed the relative abundance of proteins encoded by those genes within 10 kbp of a transcription start-site. This cohort of proteins displayed significantly reduced abundance, specifically in the H4$^{K4Q}$ mutants (Fig. 4b). Thus, proteomics analysis of trypanosomes homogeneously expressing a histone H4$^{K4Q}$ mutant suggested reduced expression of promoter-adjacent genes, without significant changes in the abundance of histones or histone variants.

### Expression of promoter-adjacent genes is disrupted in H4$^{K4Q}$ mutants

To investigate gene expression in the context of polycistronic transcription units in histone H4$^{K4Q}$ mutants in more detail, we turned to transcriptome analysis. We used the upstream borders of H4$^{K10}$ acetylation footprints to identify "transcription start-sites", and also mapped transcription termination-sites (see "Methods"). We found that *T. brucei* polycistrons contain approximately fifty genes on average, with the first and last genes located 2.7 kbp and 120 kbp on average from transcription start-sites (Fig. 5a). Consistent with the

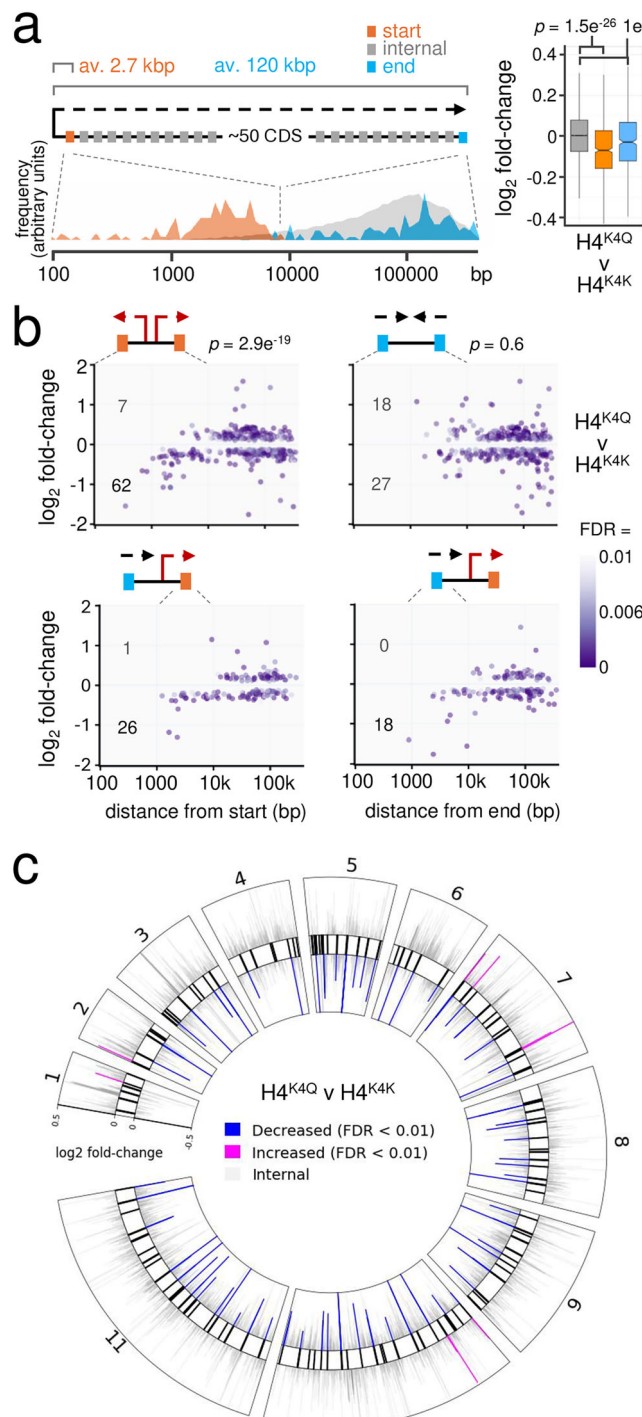

**Fig. 5 | Expression of promoter-adjacent genes is disrupted in H4$^{K4Q}$ mutants.** **a** The schematic illustrates a canonical RNA polymerase II transcribed polycistronic transcription unit in *T. brucei*, showing the distribution of genes closest to the start-site (orange), closest to a termination-site (blue) and all other genes (grey). The boxplot shows log$_2$ fold-change in RNA-seq data when we compared three strains expressing the H4$^{K4Q}$ mutant to an H4$^{K4K}$ control. Genes within 10 kbp of a transcription start-sites (orange, *n* = 736), genes within 10 kbp of a termination-site (blue, *n* = 576), all other genes (grey, *n* = 8755). Boxes indicate the interquartile range (IQR), the whiskers show the range of values within 1.5 × IQR and a horizontal line indicates the median. The notches represent the 95% confidence interval for each median. *p*-values calculated using two-sided *t*-tests. **b** RNA-seq analysis focussing on genes adjacent to divergent start-sites (top left), genes adjacent to convergent termination-sites (top right), genes adjacent to start-sites where another polycistron ends (bottom left), and genes adjacent to termination-sites where another polycistron starts (bottom right). Distances from the start or end are shown on the *x*-axis. Numbers of transcripts within 10 kbp of a start- or termination-site that were significantly (FDR < 0.01) increased or decreased in abundance in the H4$^{K4Q}$ mutant relative to the H4$^{K4K}$ control are indicated. These numbers were used to calculate *p*-values using $\chi^2$ tests. FDR, False Discovery Rate. **c** The circular plot shows the full RNA-seq dataset mapped to the *T. brucei* chromosome cores (data in grey). Transcripts from genes that are within 10 kbp of a promoter (black bars, as defined by the upstream borders of H4$^{K10}$ acetylation footprints), and that are significantly (FDR < 0.01) increased or reduced in abundance in the H4$^{K4Q}$ mutant relative to the H4$^{K4K}$ control, are highlighted.

convergent termination-sites were neither significantly reduced nor increased in abundance (Fig. 5b, *p* = 0.6, top right panel). We concluded that the expression of genes adjacent to RNA polymerase II promoters was specifically reduced in H4$^{K4Q}$ mutants.

To determine whether reduced expression was associated with transcription start-sites per se, or with additional adjacent regions, we next considered termination-sites where another polycistron starts. At these sites, transcripts from genes on both sides of the start-sites were reduced in abundance in the H4$^{K4Q}$ mutants (Fig. 5b, bottom panels). We concluded that the expression of genes on both sides of promoters was disrupted in H4$^{K4Q}$ mutants, regardless of the direction of transcription in relation to the promoter. Transcriptome-wide visualisation of these changes revealed significantly reduced expression adjacent to most promoters, with few examples of significantly increased expression adjacent to these sites (Fig. 5c). Several individual examples of these promoter-proximal regions are shown in Supplementary Fig. 4.

*T. brucei* employs RNA polymerase I to transcribe both *rRNA* and some protein-coding genes, including sub-telomeric *VSG* genes, which are expressed in a monoallelic fashion in bloodstream-form cells. We assessed the abundance of expression site associated gene (*ESAG*) transcripts derived from these polycistronic *VSG* transcription units[32] and found that expression of genes within 10 kbp of an RNA polymerase I promoter was, in contrast to genes adjacent to RNA polymerase II promoters, increased in H4$^{K4Q}$ mutants (Supplementary Fig. 5).

## Discussion

It remains unclear to what extent, and by which mechanisms, polycistronic gene expression controls rely upon chromatin in trypanosomatids. N-terminal histone tails, and tail modifications, such as lysine acetylation, play key roles in transcription control in other eukaryotes. However, trypanosomatid histone N-terminal tails, and polycistronic transcription, are highly divergent relative to the usual model eukaryotes, suggesting novel mechanisms. Many putative chromatin regulatory factors, including acetyltransferases, methyltransferases, and histone variants, are enriched in association with the unconventional promoters in *T. brucei*[9,16,33,34], but it remains unclear whether accumulation of these factors is a cause or a consequence of transcription. In addition, and as in all eukaryotes, interpretation of histone writer, reader and eraser-defective phenotypes is complicated by potential

proteomics analysis above, transcripts from genes within 10 kbp of a transcription start-site displayed significantly (*p* = 1.5e$^{-26}$) reduced abundance in H4$^{K4Q}$ mutants (Fig. 5a), as did transcripts from genes within 10 kbp of a transcription termination-site (*p* = 1e$^{-6}$). Many start-sites are situated where two polycistrons diverge and many termination-sites are situated where two polycistrons converge, while other termination-sites are situated adjacent to where another polycistron starts (see Fig. 5b). To distinguish between these regions, we first considered cohorts of genes within 10 kbp of divergent start-sites, and within 10 kbp of convergent termination-sites. Transcripts from genes adjacent to divergent start-sites were significantly reduced in abundance (*p* = 2.9e$^{-19}$) in the H4$^{K4Q}$ mutants (Fig. 5b, top left panel). In contrast, transcripts from genes adjacent to

impacts on diverse histone and non-histone substrates and binding sites.

We describe establishment of a system to directly assess the impacts of specific histone H4 residues in *T. brucei*. This required the replacement of tandem arrays of native histone *H4* genes with a single-copy ectopic and recoded *H4* gene, which was then amenable to editing. Since post-translational modifications change the properties of histones, by altering DNA-histone interactions, or by creating binding sites that recruit chromatin-interacting proteins, we focussed on lysine residues in the N-terminal tail that are known to be either acetylated or methylated.

We used site-saturation mutagenesis to profile 384 distinct mutants and isolated a panel of strains exclusively expressing mutant histones. We found that H4[K10], consistent with a role for acetylation at this site in demarcating transcriptional start sites[6], was essential for viability. In contrast, H4[K4] or H4[K14] could be replaced by a number of different amino acids. Indeed, we observed relatively few changes in gene expression in an H4[K14Q] mutant. Since acetylation on H4[K4] is depleted at promoters, in contrast to increased acetylation on other histone residues in these regions, we assessed the impact of an H4[K4Q] mutation, which mimics the acetylated state, on gene expression profiles. Both proteomic and transcriptomic analysis showed that the H4[K4Q] mutation specifically reduced the expression of genes adjacent to promoters, consistent with the view that histones and their modifications serve to focus the recruitment and action of RNA polymerase II at these sites.

Destabilised nucleosomes appear to be a conserved feature at promoters and transcription initiation sites in eukaryotes[35]. Transcription must be further coordinated at such dispersed promoters, however, and this requires a mechanism to both delimit the boundaries of initiation and also to ensure that transcription is correctly oriented. The phenotype we observe in histone H4[K4Q] mutants may reflect defects in boundary control and/or in orientation control. Defective boundary control could allow for transcription initiation over a wider region, thereby producing incomplete and poorly processed transcripts, while defective orientation control could create conflicts between RNA polymerase complexes travelling in opposite directions. Our results are consistent with the view that histone tails contribute to boundary control and/or orientation control at highly dispersed promoters in trypanosomes. Indeed, these histone tails likely contribute to the recruitment of regulatory factors and to the assembly of transcription factor hubs[17], and may also impact co-ordination with RNA processing compartments.

Histone H4 N-terminal tails are required for repressing silent mating loci in yeast[19] and we find that the expression of genes close to RNA polymerase I promoters in silent *VSG* expression sites is increased in *T. brucei* histone H4[K4Q] mutants. Although RNA polymerase I transcription is thought to initiate at all of these sites in bloodstream-form cells, transcription attenuation effectively silences all but one of these sites[36], and these silent subtelomeric polycistrons are folded into highly compact nuclear compartments[37]. The phenotype we observe could reflect defective attenuation of RNA polymerase I transcription, or inappropriate read-through of RNA polymerase II transcription[38]. Notably, inhibition of acetyl-lysine binding bromodomain factors also disrupts silencing at *VSG* expression sites[33]. Thus, histone tails are involved in controlling RNA polymerase I and RNA polymerase II polycistrons in *T. brucei*. Indeed, RNA polymerase III transcribed genes may also be affected[39].

In summary, we show that a histone H4[K4Q] mutation impacts RNA polymerase II polycistronic transcription units in *T. brucei*, specifically reducing the expression of genes adjacent to promoters. The system and approach we describe could also be used to explore how histones impact gene silencing[40], DNA replication[41], DNA recombination and repair[42], or chromosome segregation[43]. Our findings also suggest that similar approaches could be used to edit and interrogate the roles of specific residues in other genes found in tandem arrays in *T. brucei*. Our findings support the view that histone acetylation contributes to delimiting the regions where RNA polymerase II can initiate transcription in trypanosomes. We conclude that histone H4 mutagenesis provided direct evidence that histone H4 tails impact polycistronic RNA polymerase II gene expression controls, and also RNA polymerase I mediated expression controls in *T. brucei*.

## Methods

### *Trypanosoma brucei* growth and manipulation

Bloodstream form *T. brucei* Lister strain 427 (MITat 1.2), clone 221a cells, 2T1[T7-Cas9] cells[27], and derivatives were grown in HMI-11 medium at 37 °C with 5% $CO_2$. Unless otherwise stated, strains were maintained in antibiotics at the following concentrations: 1 μg/mL of hygromycin (Sigma), phleomycin, G418 or puromycin and 2 μg/mL of blasticidin (all Invivogen); selection for new transformed strains was applied at 2.5, 2, 2, 10 and 10 μg/mL, respectively. Transfections were performed using the Human T Cell Nucleofector™ Kit and Amaxa 2b device (Lonza), and programme Z-001. Ten μg of linearised plasmid DNA or PCR-product, or 40 μg of each mutagenic oligonucleotide, were typically used to transfect 25 million cells. After transfection, cells were typically grown in 50 mL of HMI-11 medium without antibiotics for 6 h and then transferred to 48-well plates with the appropriate antibiotic selection, where applicable. The pT7[sgRNA]H4 constructs were linearised using NotI prior to transfecting 2T1[T7-Cas9] cells. When replacing the native *H4* arrays (H4[NAT] sgRNA), or when editing the ectopic *H4* gene in hist[one]H4 strains (H4[ECT] sgRNA), Cas9 expression was induced by adding tetracycline (Sigma) at 1 μg/mL for 24 h prior to transfection, and tetracycline was maintained until clone or sample collection, respectively. When replacing the native *H4* arrays, transfected cultures were selected with G418 for 3–6 days, diluted and transferred to 96-well plates to obtain clonal populations. The replace.sgRNA construct was digested with BamHI and XbaI prior to transfection with hist[one]H4 cells. Puromycin resistant clones were checked for phleomycin sensitivity, to confirm correct integration. When editing the ectopic *H4* gene, transfected cultures were diluted and transferred to 96-well plates to obtain clonal populations 2–10 days after transfection. To determine growth rates, *T. brucei* cell density was measured using a haemocytometer every 24 h, and cultures were diluted to $10^5$ cells/mL in the absence of antibiotics.

### Plasmids and editing templates

Two single guide RNA (sgRNA) sequences targeting native histone *H4* genes were assembled in the pT7[sgRNA] plasmid as described[27]. These were H4[NAT]A (TTCCGCGCACATTCTCACGG) and H4[NAT]B (GAGGCGGCGGATGGAACCGC). A recoded ectopic histone *H4* gene, H4[ECT] (GenScript), was then, as a 593 bp XbaI/PstI fragment, ligated to similarly digested pT7[sgRNA], containing either the H4[NAT]A or H4[NAT]B sequence. For Cas9-driven deletion of the native *H4* arrays, a repair cassette was amplified by PCR, using the NPT-H4.F and NPT-H4.R primers (Supplementary Data 1). The resulting template incorporated a neomycin phosphotransferase (*NPT*) expression cassette flanked by 50-bp sequences that targeted regions on either side of each array for homologous recombination. To exchange the sgRNA cassette targeting native histone *H4* genes for an sgRNA cassette targeting the ectopic histone *H4* gene, we assembled the replace.sgRNA construct. First, a *PAC* cassette was amplified using the TUBrFse and PACfBst primers (Supplementary Data 1). The 1023 bp amplicon and the pT7[sgRNA] plasmid were then digested with FseI and Bst17I and ligated. The resulting construct and a fragment containing a *VSG* promoter and a *PFR2* 5′-UTR (GenScript) were then digested with NheI and BstZ17I and ligated. Finally, an H4[ECT] sgRNA sequence (GCACCTTCTTCTGCCGCTTC) targeting the ectopic histone *H4* gene was assembled in the modified pT7[sgRNA] plasmid as described[27]. All constructs derived from pT7[sgRNA]

were validated by Sanger sequencing performed using an Applied Biosystems 3730 DNA analyser. Distinct oligonucleotide repair templates (ThermoFisher; except for the degenerate ssODNs targeting K2, K17 or K18, which were from Integrated DNA Technologies) were used for site-saturation mutagenesis or to generate specific mutants, H4$^{K4H}$, H4$^{K4Q}$, H4$^{K4\Delta}$, H4$^{KI4Q}$ and H4$^{KI4\Delta}$ (Supplementary Data 1).

## *T. brucei* genomic DNA analysis

*T. brucei* genomic DNA was extracted using DNazol (ThermoFisher), following the manufacturer's instructions. A PCR assay was used to assess replacement of *H4* arrays with the *NPT* cassette. Clones were screened using the H4.out.tandem.R and H4.out.tandem.F primers (Supplementary Data 1) that annealed to regions flanking the *NPT*-editing template detailed above. Clones with correctly integrated *NPT* cassettes were then assessed using a second PCR assay with the H4.F.new and H4.R.new primers to amplify both the ectopic and any remaining native *H4* genes. The resulting PCR products were digested with SacII and separated on 2% agarose gels. To assess exchange of the H4$^{NAT}$B sgRNA with the H4$^{ECT}$ sgRNA using the replace.sgRNA construct, clones were screened using a PCR assay with the sgRNA.PAC.inc.F and sgRNA.inc.R primers (Supplementary Data 1). The expected PCR products were also checked by Sanger sequencing. All PCR reactions were performed using Q5 High-Fidelity DNA Polymerase and Q5 polymerase buffer (New England BioLabs) and a ProFlex PCR System (ThermoFisher). For Southern blotting, *T. brucei* genomic DNA (10 μg) was digested with SacII and separated on 0.8% agarose gels. Digoxigenin labelled DNA ladder (Roche) and GeneRuler 1 kb plus DNA ladder (ThermoFisher) were run in parallel. Gels were then processed using standard protocols, DNA was transferred to nylon membranes (Amersham), and UV crosslinked. A native *H4* probe was generated by PCR using the H4.F.new and H4.R.new primers (Supplementary Data 1) with wild type *T. brucei* genomic DNA as template. The product was digested with SacII, separated on an agarose gel, and the 124 bp fragment was isolated; this fragment corresponded to the 5′-end of the native *H4* gene. An ectopic *H4* probe was generated by digesting the pT7sgRNA.H4 plasmid with AflIII, separating the products on an agarose gel, and isolating the 359 bp fragment. A digoxigenin High Prime DNA Labelling and Detection Starter Kit II (Roche) was then used to label the probes, which were applied at 25 ng/mL at 45 °C (H4$^{NAT}$) or 53 °C (H4$^{ECT}$) overnight. The membrane was washed in 0.5 × SSC at 71 °C. Imaging was performed using a ChemiDoc XRS+ (BioRad) or iBright™ CL750 Imaging System. Genome sequencing for hist$^{one}$H4 and control strains was performed on a DNBseq platform (BGI); 3.5 Gbp/sample, 150 bp read length. Initial quality control of fastq files was performed using FastQC (https://www.bioinformatics.babraham.ac.uk/projects/fastqc/), followed by adapter trimming and quality filtering with Fastp (0.20.0)[44]. Processed reads were aligned to the reference genomes 427_2018 (TriTrypDB v68)[45] using Bowtie2 (2.3.5)[46] with "–very-sensitive-local" parameters. The resulting alignments were processed with SAMtools (1.9)[47] for sorting and indexing. Bam files where were transformed to bigWig track file using bamCoverage from DeepTools (3.5)[48] with –binSize 5 and –normalizeUsing RPKM. The linear coverage visualization was performed with the pyGenomeTracks[49] Python package. Bam files from parental and derived clones were analysed using bamcompare from DeepTools (3.5)[48] with –bin 200, smooth 500, –operation log2, –normalizeUsing RPKM –extendReads, –scaleFactorsMethod None, and –outFileFormat bedgraph. The circular fold-change visualization of the bedgraph files was performed with the pyCirclize (1.6) Python package (https://github.com/moshi4/pyCirclize). Specific *T. brucei* *H4* mutants were screened using *H4* PCR assays, as described above. Clones that were positive for the edited sequence and negative for the unedited sequence were confirmed by Sanger sequencing.

## RNA extraction and RT-PCR

*T. brucei* RNA was extracted using the RNeasy Mini Kit (Qiagen), following the manufacturer's instructions. RNA was reverse transcribed using a high-capacity RNA-to-cDNA kit (ThermoFisher), followed by PCR with the SL22 and MN.H4.3 primers (Supplementary Data 1); the SL22 primer is complementary to the spliced leader sequence found at the 5′ end of every *T. brucei* mRNA while the MN.H4.3 primer is complementary to the 3′ end of both the native and ectopic *H4* genes. The resulting products were digested with SacII and separated on 2% agarose gels.

## Codon-scoring following saturation mutagenesis of *H4* N-tail residues

Samples were collected 12 h, 2, 4 and 6 days after transfection with oligonucleotide editing templates. DNA was isolated and PCR assays were performed to amplify unedited *H4* (primers H4K4.cont.R and H4K4.F for K2 and K4, H4K10.cont.F and H4K10/14.R for K10, H4K14.cont.F and H4K10/14.R for K14, H4K17.18.F.cont and H4K17.18.R for K17 and K18), or edited *H4* (primers: H4K4.mut.R and H4K4.F for K2 and K4, H4K10.mut.F and H4K10/14.R for K10, H4K14.mut.F and H4K10/14.R for K14, H4K17.18.F.mut and H4K17.18.R for K17 and K18). Sequencing of edited amplicons (K2 and K4: 217 bp, K10: 220 bp, K14: 208 bp, K17 and K18: 200 bp) was performed on a BGISEQ-500 platform at BGI; 100 b paired-end reads 3.5 Gbp per sample. Oligo counting was performed with the OligoSeeker (0.0.5) Python package[50] designed to process paired FASTQ files and count occurrences of specific codons as described in ref. 51. Raw codon counts were normalized to account for differences in sequencing depth between samples by dividing each sample's counts by a correction factor (sample total/mean total across samples). For time course experiments, biological replicates were averaged at each timepoint. To visualize relative changes in codon frequency over time, counts were further normalized to the 12-h timepoint. The circular visualization of the codon data was performed with a custom Python script (3.7) using the matplolib library.

## Protein blotting

We used antibodies specific for acetylated H4$^{K4}$, non-acetylated H4$^{K4}$ [11] or acetylated H4$^{K10}$ [10]. Two million *T. brucei* cells were resuspended in LiCOR lysis buffer (137 mM Tris-HCl, 140 mM Tris base, 1% SDS, 513 μM EDTA, 7.5% glycerol, 1.2 Orange G (Invitrogen)). Samples were then treated with Pierce universal nuclease (ThermoFisher) to reduce viscosity and incubated at 70 °C for 10 min prior to loading on 12% BisTris polyacrylamide gels alongside All Blue MW standard (BioRad). MES buffer was used for electrophoresis. Proteins were transferred to nitrocellulose membranes using an iBlot 2NC stack, and iBlot 2 device (Invitrogen) set at 25 V for 7 min. Membranes were stained with Ponceau to confirm correct protein transfer and washed with PBS. The membranes were then blocked using LiCOR blocking buffer (50 mM Tris-HCl pH 7.4, 0.15 M NaCl, 0.25% bovine serum albumin, 0.05% (w/V) Tween-20, 0.05% NaN$_3$, 2% (w/V) fish scale gelatine). Membranes were incubated with primary antibodies: α-H4$^{K4}$ac (rabbit, 1:500) and mouse α-EF-1α (Millipore, 1:10,000) for 2 h at RT or with α-H4$^{K4}$non-ac (rabbit, 1:500), or α-H4$^{K10}$ac (rabbit, 1:500), and mouse α-EF-1α (1:10,000) at 4 °C overnight. Membranes were washed three times with 0.1% w/v Tween-20 in PBS and incubated with secondary antibodies: α-rabbit IRDye800 and α-mouse IRDye680 (1:15,000 and 1:10,000, respectively), for 1 h 30 min at RT; all antibodies were diluted in LiCOR blocking buffer. Membranes were then washed three times with 0.01% w/v Tween-20 in PBS, once with PBS. Blocking and washing steps were performed using a SNAPid vacuum platform, after inserting membranes in a SNAPid 2.0 miniblot holder and plastic mainframe (all Invitrogen). Blots were finally imaged using a LiCOR Odyssey CLx scanner. Images were adjusted and analysed in Image Studio ver 5.2.

## RNA-seq

RNAseq was carried out using a DNBSEQ™ platform at the Beijing Genomics Institute (BGI), 100 b paired-end reads, 30 Mbp per sample. Initial quality control of fastq files was performed using FastQC (https://www.bioinformatics.babraham.ac.uk/projects/fastqc/), followed by adapter trimming and quality filtering with Fastp (0.20.0)[44]. Processed reads were aligned to the reference genomes 427_2018 (TriTrypDB v68)[45] supplemented with the truncated set of VSG sequences and the pT7sgRNA_H4B sequence using Bowtie2 (2.3.5)[46] with "–very-sensitive-local" parameters. The resulting alignments were processed with SAMtools (1.9)[47] for sorting and indexing, and PCR duplicates were marked using Picard MarkDuplicates (2.22.3)[52]. Read counts per coding sequence were quantified using feature-Counts (1.6.4)[53] with parameters accounting for multi-mapping reads (-M) and overlapping features (-O) and configured to count only reads where both ends were mapped (-B) and to exclude chimeric reads that mapped to different chromosomes (-C). A minimum overlap fraction of 1.0 was required (–fracOverlap 1.0), ensuring that only reads completely contained within CDS features were counted. Gene features were identified using the "gene_id" attribute (-g gene_id) from the reference genome annotation file (GTF) download from TriTrypDB. For the analysis of bloodstream expression site genes, we downloaded 14 sequence and annotation files from NCBI GeneBank corresponding to the BES/TAR clones[32]. The GFF files obtained from NCBI were converted to GTF format using GFF utilities to ensure compatibility with our analytical pipeline. These expression site regions were aligned following the same strategy described previously and mapped alignments were filtered for MAPQ > 1 before counting using featureCounts. Following the alignment processes, read counts from the expression site regions, were merged with counts from the main genome. The combined dataset was then used as input for differential expression analysis using the edgeR package (3.28)[54] in R (3.6.1). We retained only genes that had a minimum count of 10 reads in at least one sample and a minimum total count of 30 reads across all samples. The cqn package (1.32) was used to compute gene length and GC content bias. The resulting offsets were incorporated into the DGEList object before computing dispersions. We fitted our data to a generalized linear model using the quasi-likelihood (QL) method via glmQLFit and we performed differential expression testing using the QL *F*-test through the glmQLFTest function. The output from this statistical test was then processed using the topTags function, which extracted the complete set of test results. We configured topTags to return all tested genes (*n* = Inf) without applying any sorting (sort.by = "none"), ensuring that the original gene order was preserved in the output. For multiple testing correction, we employed the Benjamini-Hochberg (BH) procedure (adjust.method = "BH") to control the false discovery rate. The final result_table contained the complete set of statistical results for all tested genes, including log-fold changes, *p*-values, and adjusted *p*-values (FDR).

## RNA-seq visualization

To accurately define the boundaries of polycistronic transcription units in the *T. brucei* 427 genome, we performed manual annotation of transcription start sites (TSSs) and transcription termination sites (TTSs). This manual curation process was guided by ChIP-seq footprint data for histone H4$^{K10}$ acetylation available at TriTrypDB in the 427_2018 genome[45]. The genomic distance for each gene annotated in the BES/TAR clones (ESAGs) to its respective promoter was calculated using the annotated promoter sites identified in the GFF files downloaded from GenBank. A custom Python (3.7) script was used to visualize differential expression values (log2 fold changes) with respect to the distance from TSS or TTS using the Matplotlib (3.6) and Pandas (1.4.2) Python libraries. The circular visualization was performed with the pyCirclize (1.6) Python package (https://github.com/moshi4/pyCirclize).

## Proteomics

For each *T. brucei* strain, three 50 ml cultures at $10^6$ cells/ml ($5 \times 10^7$ cells per each technical replicate) were harvested and resuspended in 5% SDS, 100 mM triethylammonium bicarbonate in water. All samples were submitted for direct data-independent acquisition mass spectrometry. Peptides (equivalent of 1.5 μg) were injected onto a nanoscale C18 reverse-phase chromatography system (UltiMate 3000 RSLC nano, Thermo Scientific) and electrosprayed into an Orbitrap Exploris 480 Mass Spectrometer (ThermoFisher). For proteomics analysis, we utilized a protein dataset derived from the *T. brucei* strain 427_2018 that matched directly to the transcript sequences used in our RNA-seq analysis. Raw files were analysed with DIA-NN (1.8.1) with C-carbamidomethylation as fixed modification, MBR option activated, the FASTA digest library-free search option activated and selection of unrelated runs activated. We extracted protein groups from the report.tsv file using the diann (1.0.1) library in R (4.2.3) with Q.Value <= 0.01 and PG.Q.Value <= 0.01. The protein groups identified as single peptide hit were considered missing values. One proteomic sample (hist$^{one}$H4 c1, technical replicate 2) exhibited an anomalously high number of missing values, likely due to technical issues during protein injection at the mass spectrometry stage. This sample was excluded from subsequent analyses. Before batch effect correction using limma removeBatchEffect function (3.54), the proteomic data was normalized by equalizing the median intensities. Missing values were imputed using the missForest (1.5) algorithm in R, which was applied separately to each set of replicated experimental conditions. The differential expression analysis was performed with the limma package using the eBayes fitting function. FDR values were computed with the toptable function in limma.

## Reporting summary

Further information on research design is available in the Nature Portfolio Reporting Summary linked to this article.

# Data availability

The genomic, transcriptomic, and amplicon-sequencing data have been deposited in the Sequence Read Archive under BioProject PRJNA1234166. The mass spectrometry proteomics data have been deposited to the ProteomeXchange Consortium via the PRIDE partner repository under accession code PXD061709. Source data are provided with this paper.

# Code availability

The OligoSeeker Python package for counting the occurrence of specific codons has been deposited in Zenodo (https://doi.org/10.5281/zenodo.15011916).

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

## Acknowledgements

We thank Nicolai Siegel (LMU, Munich) for antibodies against acetylated or non-acetylated H4$^{K4}$ and acetylated H4$^{K10}$. We thank A. Score for assistance with proteomics, G. Bravo Ruiz for assistance with proteomics analysis, and C. Onaghise for assistance with PCR optimisation. Sanger sequencing was performed by the MRC PPU DNA Sequencing and Services team. This work benefitted from the resources provided by VEuPathDB. This work was funded by a Wellcome Trust Investigator Award to D.H. (217105/Z/19/Z) and a Wellcome Trust PhD four-year Studentship award to M.N. (222326/Z/21/Z).

## Author contributions

Conceptualization; M.N., D.H. Investigation; M.N. Data curation and analysis; M.N., M.T., D.H. Supervision; J.R.C.F., D.H. Original draft; M.N., M.T., D.H. Review & editing; all authors.

## Competing interests

The authors declare no competing interests.
