## [Peer Review file · Nature Communications]

Precision-edited histone tails disrupt polycistronic gene expression controls in trypanosomes

Corresponding Author: Professor David Horn

Version 0:

Reviewer comments:

Reviewer #1

(Remarks to the Author)

In Trypanosomes, the majority of genes are transcribed sequentially from a putative promoter associated with nucleosomes containing histones that have specific post-translational modifications. The synthesized RNA is then processed to mature mRNAs by trans-splicing and polyadenylation. A number of studies have been conducted to determine the role of histone modifications in this unusual panorama. Furthermore, histone genes are present as tandem repeats with significant divergence from other eukaryotes, notably at post-translational modification locations.

In this work, Novotná et al. used cutting-edge technology to replace the tandem of histone H4 genes of the protozoan parasite *Trypanosoma brucei* in the presence of an ectopic copy of the protein. They use a mix of strategies to create parasite clones with comparable amounts of ectopic expression to endogenous genes. It was feasible to detect allowed amino acid alterations in these parasites, indicating that the H4 acetylation site of lysine 10 (H4K10) cannot be modified, implying that it is required for growth. In contrast, changes to H4K4 and H4K14 were acceptable. While alterations to this latter had no effect on protein expression in normal developing cells, changes to K4 altered RNA and protein levels for genes closer to the transcription start sites. They conclude that acetylation of histone H4's N-terminal tail impacts gene expression in *T. brucei*.

I am quite impressed with the quality and sophistication of the experiments that led to the previously mentioned conclusion. In addition to developing an excellent method for thoroughly investigating the role of several translational modifications in histones and possibly other genes, the main novelty of this work is the identification of the specific role of H4K4 acetylation in the expression of genes located near putative promoters. K10 has an important role in *Trypanosoma cruzi*, as discovered many years ago.

An important recommendation for publication is that the manuscript provide some insight into the process underlying H4K4 acetylation. Because the study monitors steady-state RNA levels rather than transcription directly, it could be the result of an indirect effect on the nuclear distribution of chromatin during RNA processing.

Below are some specific comments:

- 1- The conclusion in the summary is too vague, related to the mentioned data above. Be more specific.
- 2- In the sentence "Transcription was perturbed following depletion of *T. brucei* histone acetyltransferase 2 (HAT2), for example, but acetylation..." I would replace "but" by "and".
- 3- In the sentence "Similarly, histone H4 lysine 4 acetylation is reduced but is not eliminated in *hat3* null cells, indicating acetylation by another acetyltransferase" better, more than one acetyltransferases.
- 4- The sentence "Using an alternative approach, mutant histone H4 genes were inducibly expressed in *Trypanosoma cruzi*, but these histones contributed only 1 % of the total histone H4 pool that was incorporated into chromatin" is incorrect. High levels of H4 are incorporated and replace the normal histone.
- 5- I propose changing the last paragraph of the Introduction by posing a question and explaining how histONE provides a platform for evaluating the role of various alterations.

Reviewer #2

(Remarks to the Author)

Histone modifications play an important role in a myriad of biological processes but studying the biological roles of modifications on individual histone residues is challenging due to the high copy number of histone genes in most organisms, requiring extensive genetic modification. Furthermore, knockdown/knockout of histone-modifying enzymes can disrupt multiple histone residues simultaneously, lacking the precision of perturbation of individual modifications. Here, Novotna et al tackle this challenge by engineering *Trypanosoma brucei* strains in which over 40 native histone H4 genes are deleted and complemented with an ectopic histone H4 gene. The authors then perform saturation mutagenesis on this gene. which they then go on to perform saturation mutagenesis on. The authors identify key amino acid substitutions in histone H4 which can or cannot be tolerated and further characterize two of these mutants showing their effect on the expression of transcription start site-proximal genes.

Overall, this paper presents an impressive use of CRISPR/Cas9 for performing saturation mutagenesis on multicopy gene families in trypanosomes and provides a framework for how this can be done in the future. In addition, the paper offers unique biological insight into the importance of individual histone tail modifications in transcriptional control.

We only have a few comments mostly relating to increasing the clarity of how the experiments were performed:

Major comments

1. It was not clear to us how the ssODNs to edit the ectopic histone H4 gene were transfected; were all ssODNs transfected as a pool or were they transfected individually?
2. If ssODNs were transfected as pools, were there six sub-pools for each lysine to prevent edits on multiple residues in the same cell? This is not explicitly stated in the main text and should be clarified.
3. Do the authors know what percentage of cells were successfully edited after ssODN transfection? Presumably, for each individual H4 residue modified this percentage would be roughly equal as the same sgRNA and homology arms are used, however between different H4 residues this may have been different as different homology arms were used. We raise this point because if the percentage of edited cells was lower for one residue this would result in more rapid depletion of these cells over time.
4. We weren't entirely sure how the data in Fig 3 different from that in Figure 2C-D. Do both figures show the fitness costs of the exact same mutated histone residues (with the exception that one measures growth directly and the other is based on detection of edits by amplicon sequencing).
5. It is interesting that the H4K4Q mutation leads to reduced expression of TSS-proximal genes. Did the authors confirm that the histone H4K4Q mutant is incorporated into nucleosomes as effectively as the wild type H4? Is it possible that there is increased incorporation of histone H4.V in these cells instead of H4K4Q?
6. Do the authors believe the reduced expression of TSS-proximal genes is a direct result of the H4K4Q mutation, or does this mutation lead to lower levels of H4K10ac and/or H2A.Z at TSSs? This could be verified using H4K10ac or H2A.Z ChIP-seq. Do the authors believe that H4K4Q affects the site of transcription initiation? Kraus et al. reported a slight shift in the site of transcription initiation following HAT2 depletion. However, in their experiments, transcription shifted upstream.

Minor comments

1. Was a design tool used to design the H4NATA and H4NATB sgRNAs or were they randomly selected? If a design tool was used, this should be stated in the Materials and Methods section.
2. In Supplementary Figure 2b the "1." and "2." annotations for the PCR amplified samples should be defined in the legend (We are assuming this refers to replicates 1 and 2?).
3. Page 6 "flanked by 25 b homology arms" should be "flanked by 25 bp homology arms"
4. Why was the repair template delivered only 24 hours after DSB induction? Was this determined as the optimal time point?
5. On page 12 "Destabilised nucleosome" should be "Destabilised nucleosomes"?
6. On page 12 "accumulation of these factor" should be "accumulation of these factors".
7. In Figure 5a in the left panel the y-axis should be labelled.
8. Some of the labels in the figures are very small and difficult to read once printed.

James Budzak and Nicolai Siegel

Reviewer #3

(Remarks to the Author)

In this manuscript the authors investigated the role of histone tail-residues in the control of gene expression in *Trypanosoma brucei*. To do this they engineered *T. brucei* strains that had the histone H4 tandem arrays deleted and instead expressed a single ectopic H4 gene. These strains provided a platform for analyzing mutant histone H4 genotypes and the authors provide convincing evidence that H4 N-terminal tails control polycistronic gene expression. In general, the data are very good, but the presentation, explanation and discussion of the results are rather minimal. There is a lot of room for improvement on that front. In particular, it would be very helpful to better visualize the effects of H4^{K4Q} on promoter adjacent genes. Fig. 5 makes it very difficult to appreciate the results and actually understand them. Specific individual examples of promoter-proximal regions with genes and expression levels would be very useful. This should include the region where the effect ends. A more detailed description of what was found would also be helpful. Is the length of the affected regions similar? Any notable features when the effect ends? Does the effect end gradually? According to Fig. 5c only a fraction of promoter-proximal regions is affected by the K4Q mutation. Are there any distinguishing feature(s) for the genomic regions

that are affected vs the rest? Again highlighting a few examples would be very helpful. The analysis of termination-sites where another polycistron starts revealed that both cohorts of genes were affected. There is no discussion of this intriguing finding. Can the authors speculate what's happening with genes adjacent to the termination sites?

Minor points:

1. According to the figure legend the left blot in Fig. 1b appears to have some residual signal after stripping and re-probing. Please provide a new blot without stripping and re-probing.
2. In Fig. 2c the numbers in the middle panel need to be better aligned to the histone H4 tail sequence.
3. The authors should provide an explanation why histone H4 lysine 5 was not included in the analysis.
4. A quantification is needed for supplementary Fig. 1d.

Reviewer #4

(Remarks to the Author)

Version 1:

Reviewer comments:

Reviewer #2

(Remarks to the Author)

The authors did a great job of addressing all our comments.

Reviewer #3

(Remarks to the Author)

That authors have responded appropriately to the review. I have no further comments or concerns.

Reviewer #4

(Remarks to the Author)

Precision-edited histone tails disrupt polycistronic gene expression controls in trypanosomes

Point-by-point responses to reviewer comments

Reviewer #1:

In Trypanosomes, the majority of genes are transcribed sequentially from a putative promoter associated with nucleosomes containing histones that have specific post-translational modifications. The synthesized RNA is then processed to mature mRNAs by trans-splicing and polyadenylation. A number of studies have been conducted to determine the role of histone modifications in this unusual panorama. Furthermore, histone genes are present as tandem repeats with significant divergence from other eukaryotes, notably at post-translational modification locations.

In this work, Novotná et al. used cutting-edge technology to replace the tandem of histone H4 genes of the protozoan parasite *Trypanosoma brucei* in the presence of an ectopic copy of the protein. They use a mix of strategies to create parasite clones with comparable amounts of ectopic expression to endogenous genes. It was feasible to detect allowed amino acid alterations in these parasites, indicating that the H4 acetylation site of lysine 10 (H4K10) cannot be modified, implying that it is required for growth. In contrast, changes to H4K4 and H4K14 were acceptable. While alterations to this latter had no effect on protein expression in normal developing cells, changes to K4 altered RNA and protein levels for genes closer to the transcription start sites. They conclude that acetylation of histone H4's N-terminal tail impacts gene expression in *T. brucei*.

I am quite impressed with the quality and sophistication of the experiments that led to the previously mentioned conclusion. In addition to developing an excellent method for thoroughly investigating the role of several translational modifications in histones and possibly other genes, the main novelty of this work is the identification of the specific role of H4K4 acetylation in the expression of genes located near putative promoters. K10 has an important role in *Trypanosoma cruzi*, as discovered many years ago.

An important recommendation for publication is that the manuscript provide some insight into the process underlying H4K4 acetylation. Because the study monitors steady-state RNA levels rather than transcription directly, it could be the result of an indirect effect on the nuclear distribution of chromatin during RNA processing.

We thank the reviewer for these comments. We were happy to see that they were “quite impressed with the quality and sophistication of the experiments” and their view that we have developed “an excellent method”. We agree that histone tails may impact the distribution and function of both transcription and RNA processing machineries and have added “and may also impact co-ordination with RNA processing compartments” at the end of the paragraph in the Discussion beginning “Destabilised nucleosomes appear to be...”.

Below are some specific comments:

1- The conclusion in the summary is too vague, related to the mentioned data above. Be more specific.

R1.1: We've adjusted the final sentence in the abstract to be more specific.

2- In the sentence “Transcription was perturbed following depletion of *T. brucei* histone acetyltransferase 2 (HAT2), for example, but acetylation...” I would replace “but” by “and”.

R1.2: The point we are making here is that it has been challenging to establish links between single post-translational modifications and phenotypes of interest.

3- In the sentence “Similarly, histone H4 lysine 4 acetylation is reduced but is not eliminated in *hat3* null cells, indicating acetylation by another acetyltransferase” better, more than one acetyltransferases.

R1.3: Adjusted as suggested.

4- The sentence “Using an alternative approach, mutant histone H4 genes were inducibly expressed in *Trypanosoma cruzi*, but these histones contributed only 1 % of the total histone H4 pool that was incorporated into chromatin” is incorrect. High levels of H4 are incorporated and replace the normal histone.

R1.4: We checked the manuscript by Ramos *et al.*, 2015 again, but could not find evidence for incorporation of high levels of H4. The manuscript states “the exogenous histone corresponded to about 1% of the total amount of endogenous histone of the cells”. We’ve added a little more detail to our text here, now stating “these histones were estimated to contribute only 0.2 % or 1.4 % of the total histone H4 pool in these cells”.

5- I propose changing the last paragraph of the Introduction by posing a question and explaining how histONE provides a platform for evaluating the role of various alterations.

R1.5: Apologies but we were not entirely sure what the suggested changes were here so have not edited this paragraph.

Reviewer #2:

Histone modifications play an important role in a myriad of biological processes but studying the biological roles of modifications on individual histone residues is challenging due to the high copy number of histone genes in most organisms, requiring extensive genetic modification. Furthermore, knockdown/knockout of histone-modifying enzymes can disrupt multiple histone residues simultaneously, lacking the precision of perturbation of individual modifications. Here, Novotna et al tackle this challenge by engineering *Trypanosoma brucei* strains in which over 40 native histone H4 genes are deleted and complemented with an ectopic histone H4 gene. The authors then perform saturation mutagenesis on this gene. which they then go on to perform saturation mutagenesis on. The authors identify key amino acid substitutions in histone H4 which can or cannot be tolerated and further characterize two of these mutants showing their effect on the expression of transcription start site-proximal genes.

Overall, this paper presents an impressive use of CRISPR/Cas9 for performing saturation mutagenesis on multicopy gene families in trypanosomes and provides a framework for how this can be done in the future. In addition, the paper offers unique biological insight into the importance of individual histone tail modifications in transcriptional control.

We thank the reviewers for these comments. We were happy to see their view that our “paper presents an impressive use of CRISPR/Cas9 for performing saturation mutagenesis on multicopy gene families in trypanosomes and provides a framework for how this can be done in the future”. Also their view that our “paper offers unique biological insight into the importance of individual histone tail modifications in transcriptional control”.

We only have a few comments mostly relating to increasing the clarity of how the experiments were performed:

Major comments

1. It was not clear to us how the ssODNs to edit the ectopic histone H4 gene were transfected; were all ssODNs transfected as a pool or were they transfected individually?

R2.1: We’ve adjusted the text here to clarify: “Taking both hist^{one}H4 strains expressing the H4^{ECT} sgRNA, we induced Cas9 expression and individually delivered the three editing templates, each with a distinct randomised codon, 24 h later; thereby yielding two independent *T. brucei* libraries for each edited lysine residue”.

2. If ssODNs were transfected as pools, were there six sub-pools for each lysine to prevent edits on multiple residues in the same cell? This is not explicitly stated in the main text and should be clarified.

R2.2: We've adjusted the text here to clarify. See R2.1 above, which refers to the first three pools.

3. Do the authors know what percentage of cells were successfully edited after ssODN transfection? Presumably, for each individual H4 residue modified this percentage would be roughly equal as the same sgRNA and homology arms are used, however between different H4 residues this may have been different as different homology arms were used. We raise this point because if the percentage of edited cells was lower for one residue this would result in more rapid depletion of these cells over time.

R2.3: As clarified above, we used "two independent *T. brucei* libraries for each edited lysine residue". Editing efficiency was not assessed directly for each library. Indeed, this was not readily achievable given the expected rapid loss of viability associated with many edits. We've added more detail that reflects editing efficiency, however; "Two hundred and twenty-six clones were assessed in total, 113 following K4 editing (14 % edited), 62 following K10 editing (8% edited), and 41 following K14 editing (95% edited)" and "yielded all five of these mutants; we screened twelve sub-clones in each case". These results suggest good coverage for all 64 codons in each library (>10 million cells in each case), which is indeed supported by "broadly consistent trends for synonymous edits that encode common amino acids". We've also now added PCA plots showing reproducibility for each of the six pairs of edited libraries; "data from the duplicate libraries were pooled to generate the radial plots since these replica experiments yielded highly consistent results based on principal component analysis (Supplementary Fig. 3a)".

4. We weren't entirely sure how the data in Fig 3 different from that in Figure 2C-D. Do both figures show the fitness costs of the exact same mutated histone residues (with the exception that one measures growth directly and the other is based on detection of edits by amplicon sequencing).

R2.4: Correct. Fig. 2c shows quantification of edited sequences in pooled libraries using amplicon-seq (384 distinct edits), while Fig. 3 shows measures of growth for clones that were assessed individually (19 distinct clones and 2 controls). We've added "by directly measuring growth" to the latter text section to clarify.

5. It is interesting that the H4K4Q mutation leads to reduced expression of TSS-proximal genes. Did the authors confirm that the histone H4K4Q mutant is incorporated into nucleosomes as effectively as the wild type H4? Is it possible that there is increased incorporation of histone H4.V in these cells instead of H4K4Q?

R2.5: We did not directly assess relative incorporation of the edited histones into nucleosomes. We did assess the abundance of the core and variant histones, however, "none of which were significantly different in abundance in the H4^{K4Q} or H4^{K14Q} mutants, relative to their H4^{K4K} or H4^{K14K} counterparts". We also note that "the variant histones were approximately 30-fold less abundant on average than the core histones".

6. Do the authors believe the reduced expression of TSS-proximal genes is a direct result of the H4K4Q mutation, or does this mutation lead to lower levels of H4K10ac and/or H2A.Z at TSSs? This could be verified using H4K10ac or H2A.Z ChIP-seq. Do the authors believe that H4K4Q affects the site of transcription initiation? Kraus et al. reported a slight shift in the site of transcription initiation following HAT2 depletion. However, in their experiments, transcription shifted upstream.

R2.6: Our study was designed to establish cause v consequence rather than direct v indirect effects. So what we can say is that expression of the modified histone is a cause rather than a consequence of the polycistron expression changes we observe. The suggested ChIP-seq may reveal changes in H4K¹⁰ac or H2A.Z footprints, but we would be unable to say whether any changes we might observe were a (direct or indirect) consequence of the H4^{K4Q} edit or a (direct or indirect) consequence of the changes in gene expression. There are also many other

factors and more than fifty histone modifications that are enriched at TSSs that could be considered for assessment here. To help address this point, we've now commented in our Introduction on "more than fifty histone modifications" and added "In the case of transcription, for example, accumulation of histone modifications and chromatin-associated factors may cause transcriptional changes or be a consequence of those changes". In terms of our view on transcription initiation control, please see the paragraph in the Discussion beginning "Destabilised nucleosomes appear to be..."

Minor comments

1. Was a design tool used to design the H4NATA and H4NATB sgRNAs or were they randomly selected? If a design tool was used, this should be stated in the Materials and Methods section. The sgRNAs were manually selected.

2. In Supplementary Figure 2b the "1." and "2." annotations for the PCR amplified samples should be defined in the legend (We are assuming this refers to replicates 1 and 2?). These refer to the "two independent hist^{one}H4 strains" detailed in the main text. We've added "Data are shown for editing in both hist^{one}H4 strains expressing the H4^{ECT} sgRNA, and in the unedited parent samples, P1 and P2" to the legend to clarify.

3. Page 6 "flanked by 25 b homology arms" should be "flanked by 25 bp homology arms" 25 b is correct as this refers to single-stranded DNA templates.

4. Why was the repair template delivered only 24 hours after DSB induction? Was this determined as the optimal time point?

In a prior GFP-tagging study, we had pre-induced Cas9 expression for 3 h [PMID: 35378143]. For this study, we pre-induced for 24 h and it worked well. For H4 array replacement, we felt that the array may be initially reduced in size, which may then have facilitated replacement with the NPT template. For H4^{ECT} editing, we wondered whether the high rate of transcription by T7-polymerase may limit sgRNA accessibility. Ultimately, we were satisfied that both steps worked well with 24 h pre-induction (also see R2.3 above).

5. On page 12 "Destabilised nucleosome" should be "Destabilised nucleosomes"? Thanks for spotting this typo – now corrected.

6. On page 12 "accumulation of these factor" should be "accumulation of these factors". Thanks for spotting this typo – now corrected.

7. In Figure 5a in the left panel the y-axis should be labelled. Apologies for the omission, now labelled.

8. Some of the labels in the figures are very small and difficult to read once printed. We've increased the font sizes in Fig. 1a, 2b, 2c, and 3a.

James Budzak and Nicolai Siegel

Reviewer #3:

In this manuscript the authors investigated the role of histone tail-residues in the control of gene expression in *Trypanosoma brucei*. To do this they engineered *T. brucei* strains that had the histone H4 tandem arrays deleted and instead expressed a single ectopic H4 gene. These strains provided a platform for analyzing mutant histone H4 genotypes and the authors provide convincing evidence that H4 N-terminal tails control polycistronic gene expression. In general, the data are very good, but the presentation, explanation and discussion of the results are rather minimal. There is a lot of room for improvement on that front. In particular, it would be very helpful to better visualize the effects of H4^{K4Q} on promoter adjacent genes. Fig. 5 makes

it very difficult to appreciate the results and actually understand them. Specific individual examples of promoter-proximal regions with genes and expression levels would be very useful. This should include the region where the effect ends. A more detailed description of what was found would also be helpful. Is the length of the affected regions similar? Any notable features when the effect ends? Does the effect end gradually? According to Fig. 5c only a fraction of promoter-proximal regions is affected by the K4Q mutation. Are there any distinguishing feature(s) for the genomic regions that are affected vs the rest? Again highlighting a few examples would be very helpful. The analysis of termination-sites where another polycistron starts revealed that both cohorts of genes were affected. There is no discussion of this intriguing finding. Can the authors speculate what's happening with genes adjacent to the termination sites?

We thank the reviewer for these comments. We were happy to see their view that we “provide convincing evidence that H4 N-terminal tails control polycistronic gene expression” and that they felt that in general our “data are very good”. To address the questions here, we have now added a Supplementary Figure (S4) showing “individual examples of promoter-proximal regions”. The affected regions do align with H4K¹⁰ac footprints but note that trypanosome “promoters are characterised by regions of 5-10 kbp that are enriched for a large number of putative chromatin regulatory factors, variant histones H2A.Z and H2B.V, and more than fifty histone modifications” as detailed in our Introduction. For our interpretation of these expression patterns, please see the paragraph in the Discussion beginning “Destabilised nucleosomes appear to be...”, also dealing with ‘orientation control’ which could explain why genes adjacent to termination sites are affected.

Minor points:

1. According to the figure legend the left blot in Fig. 1b appears to have some residual signal after stripping and re-probing. Please provide a new blot without stripping and re-probing.

On reflection, we realise that this ‘residual’ signal on the Southern blot may be due to cross-reactivity, so now state “Some residual signal, or cross-reactivity, can be seen from the 736 bp band; the ectopic *H4* probe shares 84 % sequence identity with native *H4* genes”. We note that probing the same blot controls for equal loading and transfer to the membrane. We did also validate the strains using genome sequencing, which “revealed that removal of the native *H4* gene arrays on chr. 5 in the hist^{one}H4 strains was both precise and specific (Fig. 1c)”.

2. In Fig. 2c the numbers in the middle panel need to be better aligned to the histone H4 tail sequence.

Thanks for spotting this issue which was introduced during conversion to TIFF. Now corrected.

3. The authors should provide an explanation why histone H4 lysine 5 was not included in the analysis.

We selected residues that “are conserved in *Leishmania*” and have now noted that “H4^{K5} is not conserved in *Leishmania*”.

4. A quantification is needed for supplementary Fig. 1d.

Instead of quantifying these primarily qualitative RT-PCR data, we have used the RNA-seq data (Fig. 1d) to further quantify histone *H4* transcript abundance and have now replaced “97th percentile” with “80 +/-1 % relative to native *H4* transcripts”. We had already noted in the preceding section describing the recoded *H4*^{ECT} gene that “every third codon position is a G or a C, which favours increased expression”; translation in particular.

Reviewer #4:
